# Burnout and associated occupational stresses among Chinese nurses: A cross-sectional study in three hospitals

Yasira Kabakleh[1,2], Jing-ping Zhang[1]*, Mengmeng Lv[1], Juan Li[1], Silan Yang[1], Joel Swai[3,4], Hui-Yuan Li[5]

1 Xiangya school of Nursing, Central South University, Changsha, Hunan, China, 2 Department of Nursing Service Management, Aleppo University, Aleppo, Syria, 3 Department of Nephrology and Rheumatology, The Third Xiangya Hospital of Central South University, Changsha, Hunan, China, 4 Department of Nephrology, Benjamin Mkapa Hospital, Dodoma City, Dodoma Region, The United Republic of Tanzania, 5 Nether-sole School of Nursing, The Chinese University of Hong Kong, Shatin, New Territories, Hong Kong

⊕ These authors contributed equally to this work.
* jpzhang1965@csc.edu.cn

**Data Availability Statement:** All relevant data are within the manuscript and its Supporting Information files.

**Funding:** No specific funding in this study.

## Abstract

### Background

There is literature scarcity relating to burnout and other work stresses in relation to Chinese nursing-workforce performance.

### Objectives

To assess the relationship between self-esteem versus burnout; and effort-reward ratio versus favorability to the work environment, among Chinese nurses.

### Methods

We employed four validated questionnaires in the assessment burnout, self-esteem, effort-reward ratio, and favorability of nursing workplace; Maslach-Burnout Inventory (MBI), Rosenberg's self-esteem (RS), Effort-Reward imbalance (ERI) and Work-Environment Scale questionnaires (WES). Linear and ordinal regression models were utilized to assess the relationships between the variables. Analyses were conducted by using SPSS at a 95% level of significance.

### Results

We assessed 487 (Mean age: 38.8±7.1 years) nurses from three hospitals. Higher self-esteem was associated with a lower level of emotional exhaustion (Unstandardized coefficient: -0.579, p-Value<0.001); and a lower level of depersonalization (Unstandardized coefficient: -0.212, p-Value = 0.001). The relationship between self-esteem and personal achievement did not reach statistical significance. A higher effort-reward ratio was associated with less likelihood that nurses would consider their work environment favorable (Logit estimate of -0.832, p-Value = 0.014).

**Competing interests:** The authors have declared that no competing interests exist.

## Conclusion

Lower self-esteem is associated with increased burnout. A higher effort-reward ratio is associated with an enhanced perception work environment as unfavorable. We recommend psychosocial intervention programs and amendments in nursing policies to improve effort-reward imbalance among Chinese nurses.

## Introduction

The concept of occupational stress at workplaces has been an essential component in human resource management at a variety of workplaces [1]. Occupational stresses could result from work-related direct factors such as work overload, role conflicts, under promotion, poor relations with colleagues, and financial difficulties, only naming a few. These factors could be worsened by individual factors such as low self-esteem and extra-organizational factors such as family problems and life crises [2]. Occupational stress could lead to short term and bearable consequences such as irritability and fatigue to severe consequences such as job-quitting and suicide [3]. It has been shown that among many occupational stresses that affect the nursing workforce are poor material and immaterial rewards, unfavorable working conditions, and burnout [4].

Defined as a psychological syndrome, involving loss of enthusiasm for work (emotional exhaustion), a sense of pessimism (depersonalization) and a reduced sense of personal accomplishment, burnout in nursing has a prevalence ranging from as low as 10% [5], to as high as 56% [6] in different countries, with China reporting 37%. With regards to nursing finances, over 50% of nurses in Australia and over 28% in South Africa, hold multiple concurrent jobs due to reasons including seeking opportunities to increase their income [7]. In Japan, up to 60% of nurses reported being dissatisfied with their jobs due to factors, including a shortage of experienced nurses and poor relationships with physicians [6]. In Sub-Saharan Africa, not only the shortage of staff and poor working environment lead to occupation stresses among nurses but also lead to poor patient safety [8, 9]. Up to 26% of sampled nurses in Nigeria reported patient's safety measures as inadequate [8]. On the other hand, work accidents among nurses are reported to be higher among low self-esteemed nurses [10].

With increasing global concern, a number of tools have been developed and validated to measure different occupation stresses among diverse workforces, including nurses. Amid many are, Maslach Burnout Inventory to measure burnout among workers [11]; Rosenberg's self-esteem questionnaire to measure self-esteem [12]; Effort-Reward Imbalance model to measure the extent that a worker's effort tally with the amount of income [13]; and work-environment scale to estimate how much workers find their work environment favorable [14]. Despite these advances, it is still challenging to pool the global prevalence of occupation stressors due to high heterogeneity and different participants' characteristics across different countries.

To our knowledge, two studies [15, 16] have been conducted to illustrate the relationship between effort-reward imbalance and burnout, one involving global humanitarian aid workers and other involving nurses in the Netherlands. No study was found to report the relationship between effort-reward imbalance and favorability to the work environment among Chinese nurses. Moreover, no study has been conducted to illustrate the relationship between burnout and workers' self-esteem in Chinese nurses. Our study, therefore, sought to assess the

relationship between self-esteem versus burnout using; and effort-reward ratio versus favorability to the work environment, among Chinese nurses.

## Materials and methods

### Study design

This is a cross-sectional survey.

### Participants

Nurses were randomly be selected from three Chinese hospitals affiliated with Central South University, namely, First, Second, and Third Xiangya Hospitals in Hunan province, Changsha city. The inclusion criteria were: (1) Registered nurses and enrolled nurses (2) Employed full- or part-time (3) Providing direct care to patients (4) At least 1 year of nursing experience in the ward (5) Consent to participate in the study. The exclusion criteria were: (1) Student nurses, (2) Having no interest in this study (3) Believes their information will be shared with the government. Demographic details were collected from all eligible participants.

### Ethical consideration

Ethical approval by three hospitals was obtained from the Behavioral Medicine and Nursing research Ethics review board. Participants were informed about the objectives and procedures of the study before the survey, and verbal consent was obtained. Participants were guaranteed anonymity and confidential handling of their data, S1 Appendix.

### Data collection procedure

Data regarding levels of self-esteem, burnout, effort-reward ratio, and favorability of working environment were collected using four validated Chinese versions of RSES [17], MBI, WES [18], and ERI questionnaires (see S2 Appendix). Paper-based questionnaires were distributed to the wards, and head nurses made them available to all nurses. Nurses had 15–20 minutes to complete self-rating MBI-HSS, RSES, ERI, and WES questionnaires. The data were collected from January 2019 to March 2019.

### Data measurements and instruments

Four questionnaires, the Maslach Burnout Inventory-human service survey (MBI-HHS), Effort-Reward Imbalance (ERI-22), Rosenberg's self-esteem, and Work-environment scale, were used to collect the data from randomly selected nurses. These are summarized in Table 1. We obtained permission from the publisher to reproduce the questionnaires in this manuscript.

### Maslach Burnout Inventory-human services survey

This is the most valid, reliable, and widely used tool for assessment of burnout among workers. It has been translated into various languages such as Chinese by Li [19]. As firstly introduced by Maslach [20] contained three parts: emotional exhaustion with Cronbach's $\alpha = 0.912$, depersonalization with Cronbach's $\alpha = 0.842$, and personal accomplishment with Cronbach's $\alpha = 0.793$.

**Table 1. Summary of questionnaires utilized, their sub-scales and score interpretations.**

| | Work Environment Scale | | Effort-Reward Imbalance | | Maslach Burnout Inventory | | Rosenberg's Self-esteem scale | |
|---|---|---|---|---|---|---|---|---|
| | Sub-scales | Score interpretations (Mean) | Sub-scales | Score interpretations (Ratio) | Sub-scales | Score interpretations | Sub-scales | Score interpretations |
| | Management and leadership of head nurse | <2.5 Favorable; 2.5 Mixed; >2.5 Unfavorable | Effort | Effort range: 3–12; Score directly proportional to effort. | Depersonalization | <6 Low; 6–9 Normal; >9 High | Self esteem | <15 Low; 15–25 Normal; >25 High |
| | Adequate Human Resource | <2.5 Favorable; 2.5 Mixed; >2.5 Unfavorable | Reward | Reward range: 7–28; Score directly proportional to reward. | Personal Achievement | <34 Low; 34–39 Normal; >39 High | - | - |
| | Nurse-Physician Relationship | <2.5 Favorable; 2.5 Mixed; >2.5 Unfavorable | Overcommitment | Reward range: 6–24; Score directly proportional to reward | Emotional Exhaustion | <19 Low; 19–26 Normal; >26 High | - | - |
| | Nurse participation in hospital affairs | <2.5 Favorable; 2.5 Mixed; >2.5 Unfavorable | Effort-Reward ratio | < 1 More reward than effort; = 1 balanced effort and reward; >1 More effort than reward (stressed) | - | - | - | - |
| | Nursing foundation for quality of care | <2.5 Favorable; 2.5 Mixed; >2.5 Unfavorable | - | - | - | - | - | - |
| Total Score | - | | - | | 132* | | 30 | |

* Summation of 54, 30, and 48 scores for emotional exhaustion, depersonalization, and personal achievement.

## Effort—reward imbalance model 22

As firstly proposed by Siegrist [13] ERI model defines the potential mechanism of workplace stress to be the quotation between work input (i.e., effort) and gains (i.e., reward). ERI consists of three items; Extrinsic-effort has six five-leveled Likert items; Intrinsic-reward has eleven five-leveled Likert items, and overcommitment has six five-leveled Likert items. Again, the model has been translated into various languages, including Chinese, by Li [19]. The overall measure of ERI is by Effort-Reward Ratio. The internal validity of this scale was consistent, with a Cronbach's alpha of 0.78, 0.81, and 0.74 for effort, reward, and overcommitment, respectively [21]. An ERI ratio of 1 represents a balance of Effort and Reward, whereas a ratio greater than 1 reflects a disproportionate effort-reward. Individuals with an ERI ratio ≥of 1 were classified as stressed.

## Rosenberg's self-esteem questionnaire

This is one of many social-science questionnaires developed by Morris Rosenberg in 1965 utilized in measuring an individual's self-esteem. The Chinese validated translated tool is available [17] with Cronbach's alpha of 0.724 [22]. The tool is a ten-item questionnaire each of ten items consisting of four levels Likert-scales in which a person scores oneself according to their current feelings. Half number of ten items are negatively worded while the rest are positive. Total maximum score counts to 30 in a 0–3 RSES (40 in a 1–4 RSES) while 0 counts minimum in a 0–3 RSES (or 10 in a 1–4 RSES).

## Work-Environment scale

This is a 31 item assessment tool designed among five subscales, based on salient loading indicating how much favorable the nurse feels of their workplace for them to deliver their best as

possible service to the patient. The Cronbach's alpha for the Chinese version was 0.9 for the total scale and 0.87 to 0.65 for the subscales [23]. Each subscale has a coherent set of 3–10 items with four subscales containing responses to questions about the nursing image, nurses' role in the quality of patient care, staff nurse–supervisor relationships, and nurse-physician relationships. The tool is available in a number of languages, including Chinese [18].

## Variables

Our study had two (2) independent variables, i.e., self-esteem and Effort-reward Imbalance. For the descriptive purpose, Self-esteem was classified as low (if <15), normal if 15–25, and high self-esteem (if >25), while Effort-Reward Imbalance was categorized as stressed (if >1) and not stressed, (if ≤1).

On the other hand, our study had four (4) dependent variables, namely; emotional exhaustion, depersonalization, and personal achievement (i.e., defining burnout) and favorability to the work environment. For the descriptive purpose, emotional exhaustion was grouped as low (if scores <19), moderate (if scores 19–26) or high (if scores >26); depersonalization grouped as low (if scores <6), moderate (if scores 6–9) and personal achievement grouped as low (<34), moderate (34–39) and high (>39); and lastly favorability to the work environment condition was classified as favorable (if 4 or 5 sub-scores were > 2.5); mixed (if 3 or 2 sub-scores were >2.5) and unfavorable (if none sub-scores were >2.5). For analytical purposes, WES was treated as a categorical variable while other aforementioned variables as continuous.

## Bias mitigation

Bias in this study will be assessed in two levels; at study level and outcome level. Study level bias assessment will involve evaluating for selection biases and detection biases. On the outcome level, information bias and attrition bias will be assessed. To minimize reporting biases, strengthening the reporting of observational epidemiologic studies (STROBE) tool customized for cross-sectional studies will be used in the write-up of this research work. S3 Appendix.

## Sample size calculation

Using the following formula (Daniel, 1999) [24], an estimated proportion of the problem [25, 26], a total of more than 385 nurses had to be randomly selected.

## Data analysis

Data analyses were conducted depending on the objective in question using computer software, SPSS 21 (SPSS Inc., Chicago, IL, USA) at a 95% significance level. To assess the relationship between self-esteem versus emotional exhaustion, depersonalization, and personal achievement; all variables were treated as continuous variables, and the linear regression model was utilized in each.

Moreover, to assess the relationship between favorability to the work environment, versus effort-reward imbalance, the later was treated as a continuous variable while favorability to the work environment scores was treated as a categorical, ordinal variable. The ordinal logistic regression model was utilized.

# Results

## Participants

Our study involved a discussion with 846 nurses who were working in three named hospitals in China. Of 846 nurses, a total of 359 were excluded due to different reasons, including not

fulfilling inclusion criteria and fear of data being collected by the authority. All remaining 487 (Mean age: 38.8±7.1 years) were included in our study.

The sociodemographic characteristics of the participants are presented in Table 2. The majority of participants were females (91.3%), and less than half of the participants were aged (20–29) years (48.0%). Most of the participants received a bachelor's degree (68.0%), were either married (63.3%); half of them no had children (50.7.0%).

## Self esteem and burnout

Table 3 illustrates the cross-tabulation of self-esteem against burnout subscales, namely; emotional-exhaustion, depersonalization, and personal achievement. The overall mean self-esteem scores were 25.6±3.8, while overall mean scores for burnout were 18.9±4.8, 11.6±5.5 and 28.4 ±9.6 and for emotional exhaustion, depersonalization, and personal achievement. A total of 3 (0.6%), 153(31.4%), and 331(68.0%) nurses reported having low, normal, and high self-esteem, respectively.

A hundred percent of nurses who reported having high self-esteem had low and medium emotional exhaustion, with none (0%) reporting to have high emotional exhaustion. Two-thirds of nurses reported having lower self-esteem had higher levels of emotional exhaustion.

**Table 2. Demographic characteristics.**

| Characteristics | | China (Number, %) |
|---|---|---|
| Sex | Female | 445(91.3) |
| | Male | 42(8.7) |
| Age | >20 | 16(3.3) |
| | 20–29 | 234(48.0) |
| | 30–39 | 203(41.7) |
| | 40–49 | 31(6.3) |
| | 50< | 3(0.7) |
| Level education | Less than Bachelor | 115(23.6) |
| | Bachelor | 331(68.0) |
| | More than a bachelor's degree | 40(8.3) |
| Department | Internal medicine | 152(31.3) |
| | Surgery | 145(29.7) |
| | Emergency | 96(19.7) |
| | Intensive care | 94(19.3) |
| Working experience in the department (Years) | >3 years | 136(28.0) |
| | 3- >6 years | 86(17.7) |
| | 6 - >9 years | 94(19.3) |
| | 9 - >12 years | 86(17.7) |
| | 12<_years | 84(17.3) |
| Smoking Status | Yes | 39(8.0) |
| | No | 440(90.3) |
| | Sometimes | 8(1.7) |
| Alcohol intake | Yes | 52(10.7) |
| | No | 425(87.3) |
| | Sometimes | 10(2.0) |
| Exercises | Yes | 266(54.7) |
| | No | 190(39) |
| | Sometimes | 31(6.3) |

**Table 3. Frequency table of self-esteem against emotional-exhaustion, depersonalization, and personal achievement.**

| | | Self-esteem | | | | |
|---|---|---|---|---|---|---|
| | | Low (N, %) | Normal (N, %) | High (N, %) | Total (N, %) | Chi-square/p-Value |
| **Emotion Exhaustion** | Low (N, %) | 0 (0.0) | 25 (16.3) | 166 (50.1) | 191 (39.2) | 105.047/ 0.000 |
| | Medium (N, %) | 1 (33.3) | 111 (72.5) | 165 (49.9) | 277 (56.9) | |
| | High (N, %) | 2 (66.7) | 17 (11.1) | 0 (0.0) | 19 (3.9) | |
| | Total (N, %) | 3 (100) | 153 (100) | 331 (100) | 487 (100.0) | |
| **Depersonalization** | Low (N, %) | 0 (0.0) | 22 (14.4) | 50 (16.1) | 72 (14.8) | 3.823/ 0.430 |
| | Medium (N, %) | 0 (0.0) | 24 (15.7) | 70 (22.5) | 94 (19.3) | |
| | High (N, %) | 3 (100.0) | 107 (69.9) | 211 (67.8) | 321 (65.9) | |
| | Total (N, %) | 3 (100.0) | 153 (100.0) | 311 (100.0) | 487 (100.0) | |
| **Personal Achievement** | Low (N, %) | 0 (0.0) | 106 (69.3) | 208 (66.9) | 314 (64.5) | 9.455/ 0.051 |
| | Medium (N, %) | 1 (33.3) | 23 (15.0) | 69 (22.2) | 93 (19.1) | |
| | High (N, %) | 2 (66.7) | 24 (15.7) | 54 (17.4) | 80 (16.4) | |
| | Total (N, %) | 3 (100.0) | 153 (100.0) | 311 (100.0) | 487 (100.0) | |

Moreover, two-thirds of 111(72.5%) of normal self-esteem nurses reported moderate levels of emotional exhaustion.

Utilizing raw scores of emotional exhaustion and self-esteem as continuous variables, linear regression was run. Both variables were non-normally distributed. Pearson's correlation coefficient between self-esteem and emotional exhaustion was -0.458($t$ = -11.336; p-Value = 0.00); maximum and minimum standardized residuals were between −2.393 and 2.715; and Cooks distance of between 0 and 0.022. From the model, 20.8% (i.e., Adjusted $R^2$ = 0.208) of the variance in the emotional exhaustion was explained by self-esteem (p-value<0.001). The unstandardized coefficient was -0.579 (Confidence interval: −0.679−−0.479), (p-Value<0.001). This indicates that for every 1 unit of increase in self-esteem, emotional exhaustion decreases by −0.579.

Regarding depersonalization, the majority 211 (67.8%) of nurses who reported having high self-esteem had also reported higher levels of depersonalization. While a hundred percent of low self-esteem, nurses reported having higher levels of depersonalization, about two thirds (69.9%) of normal self-esteem nurses had reported having higher levels of depersonalization. Utilizing raw scores of depersonalization and self-esteem as continuous variables; linear regression was run. Both variables were non-normally distributed. The correlation coefficient of -0.146($t$ = -3.241; p-Value = 0.001); maximum and minimum standardized residuals were between -2.375 and 2.537; and Cook's distance of between 0 and 0.033. From the model, 1.9% (i.e., Adjusted $R^2$ = 0.019) of the variance in depersonalization was explained by self-esteem (p-value<0.001). The unstandardized coefficient was -0.212 (Confidence interval: −0.341−−0.083), (p-Value = 0.001). This indicates that for every 1 unit of increase in self-esteem, depersonalization decreases by 0.212.

Furthermore, the majority (66.9%) of high self-esteem and normal self-esteem (69.3%) nurses reported having lower personal achievement in contrast to two-thirds of lower self-esteem nurses who reported higher levels of personal achievements. Utilizing raw scores of personal achievement and self-esteem as continuous variables; linear regression was run. Both variables were non-normally distributed. The Pearson's correlation coefficient between the variables was 0.027($t$ = 0.593; p-Value = 0.554); maximum and minimum standardized residuals were between -1.745 and 1.778; and Cooks distance of between 0 and 0.032. From the model, 0.1% (i.e., Adjusted $R^2$ = 0.001) of the variance in personal achievement was explained by self-esteem (p-value<0.554). The unstandardized coefficient was 0.068 (Confidence

**Table 4. Frequency table of effort-reward imbalance against favorability to the work environment.**

| | | Favorability to the work environment | | | | Chi-square/p-Value |
|---|---|---|---|---|---|---|
| | | Unfavorable | Mixed | Favorable | Total | |
| Effort-Reward Imbalance | ERI<1 | 83 (23.5) | 184 (52.1) | 86 (24.4) | 353 (100.0) | 7.837/0.098 |
| | ERI = 1 | 5 (62.5) | 2 (25.0) | 1 (12.5) | 8 (100.0) | |
| | ERI>1 | 34 (27.0) | 57 (45.0) | 35 (28.0) | 126 (100.0) | |
| Total | | 122 (100.0) | 243 (100.0) | 122 (100.0) | 487 (100.0) | |

All statistics at 95% significance level.

interval: −0.157−−0.292), (p-Value = 0.554). This indicates that for every 1 unit of increase in self-esteem, personal achievement increased by 0.068. However, this finding did not reach statistical significance.

## Favorability to the work environment and effort-reward imbalance

Table 4 illustrates the frequency table of effort-reward imbalance against favorability to the work environment, as measured by ERI-22 and Work-Environment Scale, respectively. The overall mean ERI scores were 0.9±0.2, while the mean WES scores were 3.04±0.5. A total 25.1%, 49.9% and 25.1% presented with ERI-Ratio<1, ERI = 1 and ERI-Ratio>1, respectively. More than two thirds (70.5%) of nurses who thought their workplaces were favorable, had higher rewards than efforts (ERI<1). About a quarter (21.4%) of nurses who reported having mixed thoughts about the favorability of their working places also reported to be stressed (ERI>1). The majority of nurses who thought their workplaces were unfavorable had ERI<1.

Utilizing raw scores of ERI as continuous independent variables and favorability to the work environment scores as dependent categorical-ordinal variables, an ordinal logistic regression was run. From the goodness of fit results, Pearson and Deviance Chi-square values were 175.728 (p-Value = 0.597) and 188.964 (p-Value = 0.372). From the model, 0.14% of the variance in WES is explained by ERI (Pseudo $R^2$ = 0.014). The test of parallel lines was significant (p-Value = 0.007). The Logit estimate was -0.832 (p-Value = 0.014). The result signifies, the higher the effort-reward ratio, the less likely that nurses would consider their work environment favorable.

## Discussion

The idea of workplace occupational stressors has been vital in human resource management at workplaces [27, 28]. Workplace occupational stressors have been studied by many scholars and have been closely linked to a number of workers' characteristics such as psychological, i.e., low self-esteem, financial wellbeing, i.e., poor reward and incentives, as well as poor workplace environment, i.e., unfavorable [29, 30]. Following scarcity of published studies illustrating the relationship among various nursing occupational stresses among Chinese nurses, our study aimed to assess the relationship between self-esteem versus burnout using linear regression; and effort-reward ratio versus favorability to the work environment using ordinal logistic regression, among Chinese nursing workforce.

From our study, high mean self-esteem scores are illustrated by the majority of nurses 68.0% as compared to 0.6% and 31.4% who scored low and normal self-esteem, respectively. Regarding self-esteem, our study results contradict those reported by Erkorkmaz et al. [31], who reported a lower mean among Turkish nurses. The difference could be thought of due to different healthcare and employment policies that vary among countries. In another study,

Molero et al. [32] in Spain demonstrated higher proportions (41.3%) of low self-esteem, which also contradicts our results. This could be explained by the latter study involving doctors; or as a result of social desirability bias introduced in our study through fear of information being shared with the government.

Regarding burnout dimensions, higher levels of burnout sub scores have been reported in a meta-analysis by Gómez-Urquiza et al. [33]. As compared to our study, Dyrbye et al. [34] reported higher mean scores in all three subscales in the USA. Apart from differences in working departments, schedules patients' workloads and management policies that are apparent between countries, non-response, and social desirability biases could also play a role in explaining the differences as these are cross-sectional studies. In a contemporary study in China, more or less similar results were reported by Wang et al. [35] with higher mean scores of emotional exhaustion, lower mean scores of depersonalization, and lower mean scores of personal achievements. The alignment between these results could be explained by the fact that the studies were conducted in the same country that could mean the same healthcare, employment, and human resource management policies.

Regarding the relationship between self-esteem and burnout, our study revealed negative relationship with emotional exhaustion (Unstandardized coefficient: -0.212, p-Value = 0.001) and depersonalization (Unstandardized coefficient: -0.212, p-Value = 0.001); and positive relationship with personal achievement (unstandardized coefficient: 0.068, p-Value = 0.554). Supporting our study's results by proportions of burnout subscales, Erkorkmaz et al. [31] also reported a negative relationship between burnout versus self-esteem. This is due to the loss of morale for work related to loss of self-worth [36]. On the other hand, our study results contradict available contemporary literature, as demonstrated by Kupcewicz et al. [37]. Kupcewicz demonstrated that low self-esteem and part of socio-demographic variables determine burnout by studying a total of 2013 nurses. Nurses with lower self-esteem were more often experiencing symptoms of work-related burnout than personal burnout. Burnout in nurses was determined by the financial situation, the system of shift work, and education. The difference could be explained by different working conditions, healthcare policies, and differences between Poland and China. To further contradict the matter, other scholars Lou et al. [38] have demonstrated no statistically significant relationship between self-esteem and burnout in nurses.

Regarding the effort-reward imbalance, a higher effort-reward ratio was associated with less likelihood that nurses would consider their work environment favorable (Logit estimate was -0.832, p-Value = 0.014). Moreover, our study reported an overall mean score ERI that was on the higher side as compared to that reported by Colindres et al. [39] in Ecuador, a developing country. The reasons for the differences are thought to entirely be due to different nursing policies among different countries, arguing amendments in nursing management policies and the introduction of psychosocial interventions targeting effort-reward imbalance [39]. From our study, half of the nurses presented with ERI = 1, while ERI-Ratio<1 and ERI-Ratio>1 was represented by the two remaining quarters, respectively. More than two-thirds of nurses who thought their workplaces were favorable, had higher rewards than efforts (ERI<1).

About a quarter of nurses who reported having mixed thoughts about the favorability of their working places also reported to be stressed (ERI>1). This result coincides with those reported by Topa et al. [40] linking effort-reward imbalance to the creation of organizational injustice and unfavorable working environment. Unfavourability of the nursing working environment due to effort-reward imbalance has been defined differently by different authors. While Griep et al. [41] and Weyers et al. [42] describing it as one that leads to psychological stress, Lee et al. [43] defined as one leading to physical stress such as musculoskeletal disorders. Schreuder et al. [44] positively correlated effort-reward imbalance with the nursing frequency of sickness with a variety of medical conditions, including cardiovascular diseases. It follows

that an imbalanced effort-reward ratio (i.e., ERI>1) creates a sense of poor morale among workers and negatively impacts job satisfaction, performance, and turnover [45].

Despite promising results, these should be interpreted with caution due to a number of biases that were noted during the conduction process. Among several is the nuisance of four questionnaires [46]. This was a cumbersome task and perhaps dull, in that, could introduce non-response biases. Furthermore, the reliability and validity of the assessment questionnaires are not entirely perfect (i.e., 100%) [47, 48]. A large number of recruited participants were also excluded from the analysis for not meeting inclusion criteria and fear of data collection by the government. These could introduce social desirability biases, as well as making our study less representative of the actual nursing population. We call upon the robust research in the topic, especially amid the ongoing COVID-19 pandemic, mitigating biases encountered in this study.

## Conclusion

Lower self-esteem is associated with increased burnout. A higher effort-reward ratio is associated with an enhanced perception work environment as unfavorable. We recommend psychosocial intervention programs and amendments in nursing policies to improve effort-reward imbalance among Chinese nurses.

## Supporting information

**S1 Appendix. Ethical committee letter of approval.**
(PDF)

**S2 Appendix. Questionnaires.**
(PDF)

**S3 Appendix. Strengthening Reporting of Observational Epidemiologic studies (STROBE) tool: Cross-sectional studies.**
(PDF)

**S1 File.**
(ZIP)

## Acknowledgments

The author of this study is thankful for supervisory input from Dean, School of Nursing.

## Author Contributions

**Conceptualization:** Yasira Kabakleh, Mengmeng Lv, Juan Li, Silan Yang, Joel Swai, Hui-Yuan Li.

**Data curation:** Yasira Kabakleh, Mengmeng Lv, Juan Li, Silan Yang, Hui-Yuan Li.

**Formal analysis:** Yasira Kabakleh, Mengmeng Lv, Juan Li, Silan Yang, Joel Swai, Hui-Yuan Li.

**Funding acquisition:** Jing-ping Zhang.

**Investigation:** Yasira Kabakleh, Mengmeng Lv, Juan Li, Hui-Yuan Li.

**Methodology:** Yasira Kabakleh, Mengmeng Lv, Juan Li, Joel Swai, Hui-Yuan Li.

**Project administration:** Yasira Kabakleh, Jing-ping Zhang, Mengmeng Lv, Juan Li, Silan Yang, Hui-Yuan Li.

**Resources:** Yasira Kabakleh, Mengmeng Lv, Juan Li, Silan Yang, Hui-Yuan Li.

**Software:** Yasira Kabakleh, Mengmeng Lv, Juan Li, Silan Yang, Joel Swai, Hui-Yuan Li.

**Supervision:** Jing-ping Zhang.

**Validation:** Yasira Kabakleh, Jing-ping Zhang, Mengmeng Lv, Juan Li, Hui-Yuan Li.

**Visualization:** Jing-ping Zhang.

**Writing – original draft:** Yasira Kabakleh, Mengmeng Lv, Juan Li, Hui-Yuan Li.

**Writing – review & editing:** Yasira Kabakleh, Jing-ping Zhang, Mengmeng Lv, Juan Li, Hui-Yuan Li.

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
