## [Decision Letter · Decision Letter 0]

22 Jan 2020

PONE-D-19-33567

Burnout and associated occupational stresses among Chinese nurses: A cross-sectional study in three hospitals

PLOS ONE

Dear Ms Zhang,

Thank you for submitting your manuscript to PLOS ONE. After careful consideration, we feel that it has merit but does not fully meet PLOS ONE’s publication criteria as it currently stands. Therefore, we invite you to submit a revised version of the manuscript that addresses the points raised during the review process.

Your work has been reviewed by three acknowledged experts in the topic addressed in the paper. Overall, two of these reviewers remark some potential in the manuscript, but several amendments and revisions should be done before considering its acceptance in PLOS ONE. As you will find below (detailed comments provided by the reviewers), some key sections of the paper need your revision, especially those related to the methods, results and discussion of the outcomes, that need several improvements to properly support the conclusions of the paper.

Please refer to ALL the comments provided by the reviewers, detailing your responses and rationales at the rebuttal letter, and tracking the changes performed in one of the two copies of the revised manuscript to be resumbmitted.

We would appreciate receiving your revised manuscript by Mar 07 2020 11:59PM. To enhance the reproducibility of your results, we recommend that if applicable you deposit your laboratory protocols in protocols.io, where a protocol can be assigned its own identifier (DOI) such that it can be cited independently in the future. For instructions see: http://journals.plos.org/plosone/s/submission-guidelines#loc-laboratory-protocols

We look forward to receiving your revised manuscript.

Kind regards,

Sergio A. Useche, Ph.D.

Academic Editor

PLOS ONE

Journal Requirements:

2. Please clarify in your Methods section whether the questionnaires are published under a CC-BY license, or whether you obtained permission from the publisher to reproduce the questionnaires in this manuscript. Please explain any copyright or restrictions on this questionnaire.

4. Thank you for stating the following in your Competing Interests section:  "None"

7. Please upload a copy of Supporting Information Table 1-4 which you refer to in your text on page 19.

Reviewers' comments:

Reviewer's Responses to Questions

**Comments to the Author**

1. Is the manuscript technically sound, and do the data support the conclusions?

Reviewer #1: Partly

Reviewer #2: Partly

Reviewer #3: No

2. Has the statistical analysis been performed appropriately and rigorously? 

Reviewer #1: Yes

Reviewer #2: Yes

Reviewer #3: No

3. Have the authors made all data underlying the findings in their manuscript fully available?

Reviewer #1: Yes

Reviewer #2: Yes

Reviewer #3: No

4. Is the manuscript presented in an intelligible fashion and written in standard English?

Reviewer #1: Yes

Reviewer #2: Yes

Reviewer #3: No

5. Review Comments to the Author

Reviewer #1: I found the background very informative. I appreciate the international scope of the problem. The paper does a good job using published work to build a case for the proposed study.

I suspect that most readers will not be familiar with the details of the 1) Maslach Burnout Inventory-human service survey (MBI-HHS), 2) Effort-153 Reward Imbalance (ERI-22), 3) Rosenberg’s self-esteem and 4) Practice environment scale. A table that list the major factors addressed in most of these scales would help the readers better interpret the results. The factors should be grouped by category so that it is apparent how the scales complement each other. Also, the table should include ranges and normal values for these metrics to help the reader interpret the reported values in your study.

The analysis was based on a linear regression and logistic regression (line 311). I'd really like to see some scatter plots of the data in which the ranges on the horizontal and vertical axes are based on the theoretical minimum and maximums for workers.

I have some concern about the underlying statistical validity of the study design and the statistical methodology. While random selection procedures were used, subject selection was not balanced on factors hypothesized apriori to affect burnout, e.g., age, gender, years of work, education, speciality, income, etc. If you don't have significant differences in you independent variables, you probably won't have sufficient statistical power to find significant differences in your dependent variations. Also, there is a problem with natural selection that can bias the findings, A.K.A. "health worker effect." (a problem in all cross-sectional studies) This does not mean that your findings are not important, but it does require a careful discussion of study design limitations and there affect on the results.

The discussion could strengthen with more cross-referencing of tables & figures that contain supporting data. Also, it would be helpful if you indicated the possible range for the variables you are discussion. Fore example (lines 341-345) "Nurses with lower self-esteem were more often experience symptoms of work-related burnout (r=-0.21; p<0.001) and in nurses (r=-0.20; p<0.001), than personal burnout (r=-0.14; p<0.0001). Burnout in nurses was determined by the financial situation (p<0.001), the system of shift work (p<0.0001) and education (p<0.0001)." First of all, these are really small correlations -- irrespective of their "p" values. This may be related to a lack of variability in your independent variables. Second, I really don' have a clear idea of what is "lower." Can you say 2 versus 5 on a scale of 0 to 10?

Since this is a correlation study, consider adding a correlation matrix with all of the the independent and dependent variables. I'd like to see the correlations among the dependent variables and among the independent variables.

The main conclusion appears to be "For nursing workforce to consider their workplaces favorable and offer best performances, nurses’ psychological wellbeing and financial reward need to be considered as a whole-pair of factors and not as separate." I don't doubt that this is important, but I felt let down that you did not include other important findings. For example, in lines 311-382, you talk about burnout and self esteem scores in your and others' studies, but I don't get a clear idea of: Are the overall levels of burnout and self esteem in your study group high and low respectively? Are they substantially higher and lower than populations in other countries? Are they substantially higher or lower than other occupational groups, etc. These findings are important and should be summarized in the conclusions.

Reviewer #2: The theme of the manuscript is of relevant importance, because Burnout Syndrome is present in several professional categories, nurses one of the most affected.

1. The manuscript describes technically sound piece of scientific research with data that supports the conclusions. but the finalization of the research was not clear. In the final outcome, the exact correlation between well-being versus burnout, or effort-reward ratio versus work-environment (research objectives) is unclear.

2. The statistical analysis is very detailed.

3. In the background, the author reports that there is a scarce literature relating burnout and other work stressors in Chinese nurses. Therefore, the author could suggest further investigations on the subject.

4. At line 13, number 4 is not in standard format.

5. Krause, N., R. Rugulies, and C. Maslach is in the References number 7, but was not cited in the main text.

6. Reference number 22 is in invalid format.

Reviewer #3: The relationship between burnout and occupational stressors in nurses is definitely a fascinating area to look into. However, the manuscript presents a number of shortcomings that will need to be addressed.

1. Abstract. Length. This abstract is perhaps too long (375 words). The longest - and perhaps ambiguous section - is the ‘Results’ section. I would recommend revising this section by shortening the amount of information provided (e.g. reporting the most relevant results).

2. Abstract. Objectives. The authors use the word ‘correlation’ throughout the manuscript when describing regressions. This is concerning given the fact that these are two different statistical tests and analyses. Perhaps the authors’ intention was the use of ‘relationship’ instead?

3. Abstract. Results. There is contradictory information reported (e.g. “The majority 211 (67.8%) of nurses who reported to have high self-esteem had higher levels of depersonalization. The higher the self-esteem the lower level of depersonalization [correlation coefficient of -0.146(t=-3.241; p- Value=0.001)]). Please revise.

4. Introduction. English. The English language employed lacks clarity in several sections of the manuscript (e.g. ‘In regard to burnout, despite trending in early nineteen-eighties, the condition was previously regarded as work-related depression [1, 2].)’, contains incorrect or inappropriate words ‘life of style’ or ‘for ages’, lack of references (‘e.g. Loss of morale for work generally defines burnout and could significantly and negatively impair employee’s health, organization, performance and productivity.). I would recommend re-writing.

5. Introduction. Content. The authors include a rationale for the correlation of the ERI, burnout and self-esteem but in the actual analyses they do not report any correlations. Also, It would be good to include a strongest rationale for the second research question.

6. Methods. Ethics. Specification about the ethics statement (Was it granted from a University?).

6. Methods. Aims. The authors report the word ‘association’ when talking about a regression. This leads to confusion given the fact that we usually refer to the association in the context of regressions. I recommend re-wording and reporting a correlation table.

7. Methods. Data measurements and instruments. There should be a section where the authors describe the validity and reliability of the measures employed. Information regarding the scoring should be shortened.

8. Methods. Exclusion criteria. The authors excluded a large number of participants (n=359) in this study. However, it concerns that the only exclusion criteria mentioned are: 1) Student nurses; 2) Having no interest in this study. Considering the overall sample size and the number of participants excluded, how did you measure “no interest in the study”?

9. Methods. Sample size calculation. It is good to see the authors reported a calculation. However, this could be mentioned, shortened and inserted ‘in-text’. There is no need to report a sample size formula in the manuscript.

10. Results. Tables. Descriptive stats. I would shorten Table 1 and only include relevant information. Indeed, this table also include not relevant information.

11. Results. Table 3. The Table does not present any information regarding the description of the beta coefficient, indicates “correlation” in the title when should read ’regression’, and does not include the level of significance of the p value.

12. Results. Description of results and tables. Each table should include notes regarding the level of significance.

13. Discussion. The references of the studies are reported in two formats. Please revise and delete the APA format. Indeed, PLOS one accepts manuscript with X referencing style.

14. Discussion. There is no need to report information presented elsewhere in the manuscript (e.g. means, SDs, etc..) - for instance, in the results section. Moreover, when comparing the studies, there is no need to report exact information from other studies (e.g. means and SDs).

15. Discussion. I would include as a limitation the fact that the authors excluded a large number of nurses from the previously recruited.

6. PLOS authors have the option to publish the peer review history of their article (what does this mean?). If published, this will include your full peer review and any attached files.

Reviewer #1: No

Reviewer #2: No

Reviewer #3: No

---

## [Author Response · Author response to Decision Letter 0]

13 Feb 2020

Postal Code 410013,

Hunan Province, P.R. China

10th February, 2020.

Editor-in-Chief,

PLOS ONE- Journal 

jheber@plos.org

San Francisco, USA

Dear Sergio A. Useche,

RE: REVISED MANUSCRIPT (PONE-D-19-33567) SUBMISSION

Please refer to the heading above. Thank you for your prompt responses.

I am Yasira Kabakleh the first author for manuscript (PONE-D-19-33567) entitled; ‘Burnout and associated occupational stresses among Chinese nurses: A cross-sectional study in three hospitals. 

Regarding data availability, there is no ethical or legal restrictions on sharing the data. We have therefore included our data in the revised submission.

Regarding ORCID, we have validated the ORCID.

Regarding supporting information Table 1-4 which we referred in page 19, we only meant tables that are embedded in the manuscripts. i.e. Table 1 – 5. 

Regarding questionnaire’s copyright, authors obtained permission from the publisher to reproduce the questionnaires in this manuscript.

Regarding competing interests, the authors have declared that no competing interests exist.

Below are point-to-point responses to your reviewers’ comments. 

Reviewer #1: 

I found the background very informative. I appreciate the international scope of the problem. The paper does a good job using published work to build a case for the proposed study. Authors’ response: Noted!

I suspect that most readers will not be familiar with the details of the 1) Maslach Burnout Inventory-human service survey (MBI-HHS), 2) Effort-153 Reward Imbalance (ERI-22), 3) Rosenberg’s self-esteem and 4) Practice environment scale. A table that list the major factors addressed in most of these scales would help the readers better interpret the results. The factors should be grouped by category so that it is apparent how the scales complement each other. Also, the table should include ranges and normal values for these metrics to help the reader interpret the reported values in your study. Authors’ response: Noted! We added a new table and cited it as Table 1.

The analysis was based on a linear regression and logistic regression (line 311). I'd really like to see some scatter plots of the data in which the ranges on the horizontal and vertical axes are based on the theoretical minimum and maximums for workers. Authors’ response: Agreed! We have added scatter plots as two separate files named. Objective_One_SPSS_Output_File and Objective_Two_SPSS_Output_File.

I have some concern about the underlying statistical validity of the study design and the statistical methodology. While random selection procedures were used, subject selection was not balanced on factors hypothesized apriori to affect burnout, e.g., age, gender, years of work, education, speciality, income, etc. If you don't have significant differences in you independent variables, you probably won't have sufficient statistical power to find significant differences in your dependent variations. Also, there is a problem with natural selection that can bias the findings, A.K.A. "health worker effect." (a problem in all cross-sectional studies) This does not mean that your findings are not important, but it does require a careful discussion of study design limitations and their effect on the results. Authors response: Age, gender, years of work, education, speciality, income, etc were only collected as demographic characteristic and were not included in the analysis. We added ‘health worker effect’ In the bias.

The discussion could strengthen with more cross-referencing of tables & figures that contain supporting data. Also, it would be helpful if you indicated the possible range for the variables you are discussion. Fore example (lines 341-345) "Nurses with lower self-esteem were more often experience symptoms of work-related burnout (r=-0.21; p<0.001) and in nurses (r=-0.20; p<0.001), than personal burnout (r=-0.14; p<0.0001). Burnout in nurses was determined by the financial situation (p<0.001), the system of shift work (p<0.0001) and education (p<0.0001)." First of all, these are really small correlations -- irrespective of their "p" values. This may be related to a lack of variability in your independent variables. Second, I really don' have a clear idea of what is "lower." Can you say 2 versus 5 on a scale of 0 to 10? Authors’ response: We rephrased the discussion and conclusion sections. Regarding ranges of values (i.e. ‘lower’), we added a table interpreting lower, normal and higher scores. 

Since this is a correlation study, consider adding a correlation matrix with all of the independent and dependent variables. I'd like to see the correlations among the dependent variables and among the independent variables. Authors’ response: We mistakenly used the word ‘correlation’ in place of ‘regression’. Our study was not aimed at conducting correlations. 

The main conclusion appears to be "For nursing workforce to consider their workplaces favorable and offer best performances, nurses’ psychological wellbeing and financial reward need to be considered as a whole-pair of factors and not as separate." I don't doubt that this is important, but I felt let down that you did not include other important findings. For example, in lines 311-382, you talk about burnout and self-esteem scores in your and others' studies, but I don't get a clear idea of: Are the overall levels of burnout and self-esteem in your study group high and low respectively? Are they substantially higher and lower than populations in other countries? Are they substantially higher or lower than other occupational groups, etc. These findings are important and should be summarized in the conclusions. Authors’ response: Agreed! We rephrased the discussion part and conclusion sections accordingly.

Reviewer #2: 

The theme of the manuscript is of relevant importance, because Burnout Syndrome is present in several professional categories, nurses one of the most affected.

1. The manuscript describes technically sound piece of scientific research with data that supports the conclusions. but the finalization of the research was not clear. In the final outcome, the exact correlation between well-being versus burnout, or effort-reward ratio versus work-environment (research objectives) is unclear. Author’s response: Agreed! We rephrased the conclusion section.

2. The statistical analysis is very detailed. Author’s response: Noted!

3. In the background, the author reports that there is a scarce literature relating burnout and other work stressors in Chinese nurses. Therefore, the author could suggest further investigations on the subject. Author’s response: Agreed! We added a statement of recommendation in the discussion part.

4. At line 13, number 4 is not in standard format. Author’s response: Agreed! We edited number 4 into superscript.

5. Krause, N., R. Rugulies, and C. Maslach is in the References number 7, but was not cited in the main text. Authors response: In the paragraph two of the introduction, reference 7 was embedded among three references cited i.e. [6-8]

6. Reference number 22 is in invalid format. Author’s response: Agreed! we corrected the reference.

Reviewer #3: 

The relationship between burnout and occupational stressors in nurses is definitely a fascinating area to look into. However, the manuscript presents a number of shortcomings that will need to be addressed.

1. Abstract. Length. This abstract is perhaps too long (375 words). The longest - and perhaps ambiguous section - is the ‘Results’ section. I would recommend revising this section by shortening the amount of information provided (e.g. reporting the most relevant results). Author’s response: Agreed! We cut down the number of words in the abstract and rephrased ‘’results’’ section.

2. Abstract. Objectives. The authors use the word ‘correlation’ throughout the manuscript when describing regressions. This is concerning given the fact that these are two different statistical tests and analyses. Perhaps the authors’ intention was the use of ‘relationship’ instead?). Author’s response: Agreed! We indeed intended to mean ‘relationship’ instead of ‘correlation’. We replaced accordingly.

3. Abstract. Results. There is contradictory information reported (e.g. “The majority 211 (67.8%) of nurses who reported to have high self-esteem had higher levels of depersonalization. The higher the self-esteem the lower level of depersonalization [correlation coefficient of -0.146(t=-3.241; p- Value=0.001)]). Please revise. Author’s response: Agreed! We have revised the matter.

4. Introduction. English. The English language employed lacks clarity in several sections of the manuscript (e.g. ‘In regard to burnout, despite trending in early nineteen-eighties, the condition was previously regarded as work-related depression [1, 2].)’, contains incorrect or inappropriate words ‘life of style’ or ‘for ages’, lack of references (‘e.g. Loss of morale for work generally defines burnout and could significantly and negatively impair employee’s health, organization, performance and productivity.). I would recommend re-writing. Author’s response: Agreed! We corrected the matter by rewriting.

5. Introduction. Content. The authors include a rationale for the correlation of the ERI, burnout and self-esteem but in the actual analyses they do not report any correlations. Also, it would be good to include a strongest rationale for the second research question. Author’s response: Agreed! We mistakenly used ‘correlation’ while meaning ‘regression’. We corrected the issue and added stronger rationale for the second research question. 

6. Methods. Ethics. Specification about the ethics statement (Was it granted from a University?). Author’s response: It was granted by the Behavioral Medicine and Nursing research Ethics review board of Central south university. The three hospitals from which data were collected are all university hospitals affiliated with Central South University.

6. Methods. Aims. The authors report the word ‘association’ when talking about a regression. This leads to confusion given the fact that we usually refer to the association in the context of regressions. I recommend re-wording and reporting a correlation table. Author’s response: We rephrased the areas with association-regression confusion. We however did not report a correlation table because what we meant was ‘regression’ and not ‘correlation’. 

7. Methods. Data measurements and instruments. There should be a section where the authors describe the validity and reliability of the measures employed. Information regarding the scoring should be shortened. Author’s response: Agreed! We have added Cronbach’s alpha for each questionnaire used. We rephrased MBI-HSS paragraph to shorten it.

8. Methods. Exclusion criteria. The authors excluded a large number of participants (n=359) in this study. However, it concerns that the only exclusion criteria mentioned are: 1) Student nurses; 2) Having no interest in this study. Considering the overall sample size and the number of participants excluded, how did you measure “no interest in the study”? Author’s response: Agreed! The ‘’no interest in the study’’ was mainly due to the fear that information was collected by the government. Only those nurses who had not worried about this were included as they would be non-response bias. 

9. Methods. Sample size calculation. It is good to see the authors reported a calculation. However, this could be mentioned, shortened and inserted ‘in-text’. There is no need to report a sample size formula in the manuscript. Author’s response: Agreed! We rephrased the sample size section; shortening it by removing the formula. 

10. Results. Tables. Descriptive stats. I would shorten Table 1 and only include relevant information. Indeed, this table also include not relevant information. Author’s response: Agreed! We shortened the table trying to eliminate irrelevant information.

11. Results. Table 3. The Table does not present any information regarding the description of the beta coefficient, indicates “correlation” in the title when should read ’regression’, and does not include the level of significance of the p value. Author’s response: Agreed! We replaced ‘correlation’ with ‘regression’. All statistics were done at 95% confidence interval. We stated this in the abstract and on each table. We have attached scatter plots and two full SPSS output files named. Objective_One_SPSS_Output_File and Objective_Two_SPSS_Output_File.

12. Results. Description of results and tables. Each table should include notes regarding the level of significance. Author’s response: Agreed! All statistics were done at 95% confidence interval. We stated this in the abstract and on each table.

13. Discussion. The references of the studies are reported in two formats. Please revise and delete the APA format. Indeed, PLOS one accepts manuscript with X referencing style. Author’s response: Agreed! We downloaded and used PLoS One recommended style from https://endnote.com/style_download/plos-public-library-of-science-all-journals/ .

14. Discussion. There is no need to report information presented elsewhere in the manuscript (e.g. means, SDs, etc..) - for instance, in the results section. Moreover, when comparing the studies, there is no need to report exact information from other studies (e.g. means and SDs). Author’s response: Agreed! We rephrased the discussion part accordingly.

15. Discussion. I would include as a limitation the fact that the authors excluded a large number of nurses from the previously recruited. Author’s response: Agreed! We edited accordingly.

Authors are indeed grateful for your time reviewing our manuscript and promptly respond. Kindly, review our manuscript and hopefully publish. I am looking forward hearing from you soon. 

Best regards,

Yasira Kabakleh, RN.

First Author

Xiangya School of Nusring

Central South University

Postal Code 410013,

Hunan Province, P.R. China

Email: yasiraghazi@gmail.com

---

## [Decision Letter · Decision Letter 1]

6 May 2020

PONE-D-19-33567R1

Burnout and associated occupational stresses among Chinese nurses: A cross-sectional study in three hospitals

PLOS ONE

Dear Prof. Zhang,

Thank you for submitting your manuscript to PLOS ONE. After careful consideration, we feel that it has merit but does not fully meet PLOS ONE’s publication criteria as it currently stands. Therefore, we invite you to submit a revised version of the manuscript that addresses the points raised during the review process.

The revisions you have done in the past stage are good, but more work is needed to consider the paper as publishable in PLOS ONE. Below, you will find several concerns, questions and suggestions provided by our reviewers, in regard to various key aspects of the paper. Please focus on the support given to the conclusions, that seems to exceed the actual scope of the data, and improve the background, discussion and limitations according to the reviewer's comments. Also, our reviewers have largely problematized on the design, the nature, complexity and interpretation of the statistical analyses performed and the proper presentation of these results. However, all the comments raised by the reviewers must be clearly responded and tracked in one of the versions of the paper to be submitted.

We would appreciate receiving your revised manuscript by Jun 20 2020 11:59PM. To enhance the reproducibility of your results, we recommend that if applicable you deposit your laboratory protocols in protocols.io, where a protocol can be assigned its own identifier (DOI) such that it can be cited independently in the future. For instructions see: http://journals.plos.org/plosone/s/submission-guidelines#loc-laboratory-protocols

We look forward to receiving your revised manuscript.

Kind regards,

Sergio A. Useche, Ph.D.

Academic Editor

PLOS ONE

Reviewers' comments:

Reviewer's Responses to Questions

**Comments to the Author**

1. If the authors have adequately addressed your comments raised in a previous round of review and you feel that this manuscript is now acceptable for publication, you may indicate that here to bypass the “Comments to the Author” section, enter your conflict of interest statement in the “Confidential to Editor” section, and submit your "Accept" recommendation.

Reviewer #1: All comments have been addressed

Reviewer #2: (No Response)

Reviewer #3: (No Response)

2. Is the manuscript technically sound, and do the data support the conclusions?

Reviewer #1: Partly

Reviewer #2: Partly

Reviewer #3: No

3. Has the statistical analysis been performed appropriately and rigorously? 

Reviewer #1: Yes

Reviewer #2: Yes

Reviewer #3: No

4. Have the authors made all data underlying the findings in their manuscript fully available?

Reviewer #1: Yes

Reviewer #2: Yes

Reviewer #3: Yes

5. Is the manuscript presented in an intelligible fashion and written in standard English?

Reviewer #1: (No Response)

Reviewer #2: Yes

Reviewer #3: No

6. Review Comments to the Author

Reviewer #1: I appreciate the authors responses to my request to for clarification regarding the survey tools and data presentation. I still have some problems with the concerns I mentioned in my previous review.

The authors responded that "We mistakenly used the word ‘correlation’ in place of ‘regression’. Our study was not aimed at conducting correlations." Correlations and regressions are based on similar math. More importantly this is a cross sectional study based on a population of convenience. The sample was not balanced on key factors such as gender, age, seniority, medical specialty, etc. As such it is subject to survival biases and multicollinearity among the study variables. Cross-sectional study can show associations among variables, but they don't establsh causation. This does not mean that such studies are not useful for building models, but it is important to consider all of the ways that variables not fully controlled in the study can misinform the user. These factors merit a careful discuss and a carefully worded conclusion. I don't believe the discussion between lines 358 and 366 really do this justice. Also, the oblique reference to "loop holes" does little to inform the reader.

Finally I find the conclusions overreaching. " We believe that ... we encourage ... We encourage... We also call..." This is a call for action -- not a conclusion. This is a scientific paper -- please stick to the facts.

Reviewer #2: Burnout and occupational stress is a very important topic these days. Studies on the relationship between self steem and Burnout is very interestingto prevent this syndrome in nurses. Some points were not clear in the research and can be improved.

1- The conclusion is very small and it was not clear whether the objectives have been achieved. If the objectives was achieved or not it has be on course completion.

2- Krause, N., R. Rugulies, and C. Maslach They are in the references, but are not mentioned in the main text. Please, Check it.

3- The authors report that there is little research on this topic, but do not suggest further studies on the subject. It is important to suggest to readers new research on the topic, becouse the authors reports there are few studies on the subject.

Reviewer #3: Thank you for your responses to my review comments. I believe your study is interesting. However, I still have major concerns related to the current version of this manuscript. Therefore, I would like to suggest some further considerations, revisions and major points of reflections:

Main concerning points:

• To test the relationship between self-esteem and burnout, the authors report having conducted only regression analyses in their study. A correlation matrix is not reported or mentioned. A correlation test is essential for running regressions.

• The authors report the regressions with a ‘correlation coefficient’, and regressions are presented in the form of correlations ‘the higher the self-esteem, the lower level of emotional exhaustion’. This is incorrect and may lead to think that regressions have been conducted without running correlations analyses. If the authors aimed to test the association between these two variables, then they should re-run the analyses because the statistical test selected is not correct.

• Another primary concern comes from the number of participants excluded. In one of the points under review, the authors suggested that a large number of participants was excluded ‘due to fear that information was collected by government’. This is an important point that should be included in the study. Indeed, the fact that a large percentage of the recruited participants decided not to take part may lead to bias (e.g. social desirability?). Also, it seems that the percentage of nurses reporting higher self-esteem is much higher than in higher countries. May this be related to, again, this bias?

• The manuscript needs proof-reading throughout. There are still severe shortcomings (e.g. two styles of referencing style in the Discussion).

Previous points

#1. Abstract. Length. In my opinion, this abstract remains still pretty long. I would recommend the following changes.

-line 26: remove ‘using linear regression’ and ‘using ordinal logistic regression the objectives as these are part of the method of analyses

-line 32: remove ‘ Demographic characteristics were recorded and descriptive statistics were illustrated.’

-line 35: change ‘by SPSS’ to ‘by utilizing SPSS’

-line 52-54: remove ‘we also.. to favorability’.

- the conclusion section could be further shortened

#8. Methods. Exclusion criteria. Thank you for clarifying that issue related to the exclusion criteria. However, if the nurses agreed to take part only if they did not have fear to be fired by the government, I reckon that your results may suffer from social desirability bias? I still think you should add more specification to the exclusion criteria in the Method section. Also, I would add this point as a limitation in the Discussion.

New points

2. Introduction. I am glad the authors revised some sections. However, I think that the Introduction lacks a good and strong rationale, especially for self-esteem, effort-reward and work environment. There is a mention of the rationale for burnout but not for the other variables. Also, every construct needs an introduction. Therefore, I would suggest the following structure: 1) introduction of the construct; 2) definition, 3) rationale, 4) description of the Measures. In the current version of the manuscript, the authors defined burnout, then introduced the authors' variables and came back to the definition of burnout with a definition. I would recommend changes in this structure. Also, it would be good to mention more supporting literature.

2b. Introduction. Line 79, you mentioned a large cross-sectional study, there is no need to mention the percentage of every country.

3. Participants. From Line 127 ‘Furthermore...’ you include information related to the Procedure and not the Participants of the study. This info should be placed either into a new section named ‘Procedure’.

4. Data measurements and Instruments. I would place this section before variables. I would include information about the Variables under the description of each questionnaire in the ‘Data measurements and Instruments’.

5. Analyses. Correlations. The authors report the regressions but do not report the correlation matrix. I don’t think it is acceptable to undertake regressions without running and reporting a correlation matrix. I strongly encourage the authors to do so.

6. Appendix, Tables and Figures. There are a lot of Appendix, Tables and Figures in the manuscript. First, I would place all the questionnaires into one Appendix. Second, I would reduce the number of Tables and Figures. Information placed in-text does not necessarily need to be repeated elsewhere.

7. Ethical consideration. This information is usually placed at the start of the Method section (e.g. under ‘Participants’).

8. Results. Information placed under the Descriptive Data regarding nurses should go under Participants.

9. The ‘work environment’ construct may be easily confused with work stressors (organizational stressors). I would recommend referring to this variable as ‘favourability to the work environment (context)’.

10. Discussion. References. The Discussion section still includes two types of referencing styles. The authors should only include ‘ Vancouver’ as the preferred referencing style.

11. Discussion. I am pretty impressed to read that 68% of the nurses have high self-esteem. Is this related to the social desirability bias and the fact that nurses felt the pressure from the government to respond? If you think that may be the case, it would be good to mention it.

I hope these comments will help the authors to improve their manuscript.

7. PLOS authors have the option to publish the peer review history of their article (what does this mean?). If published, this will include your full peer review and any attached files.

Reviewer #1: No

Reviewer #2: No

Reviewer #3: No

---

## [Author Response · Author response to Decision Letter 1]

23 Jun 2020

Postal Code 410013,

Hunan Province, P.R. China

23rd June, 2020.

Editor-in-Chief,

PLOS ONE- Journal 

jheber@plos.org

San Francisco, USA

Dear Sir/Madam

RE: RE: REVISED MANUSCRIPT (PONE-D-19-33567) SUBMISSION

Please refer to the heading above. I am Yasira Kabakleh a PhD student in Nursing at Central South university, in China. I am the first author of the manuscript titled; “Burnout and associated occupational stresses among Chinese nurses: A cross-sectional study in three hospitals” 

Below are point-by-point authors’ responses to the points raised by the reviewers. 

Reviewer #1: 

1) I appreciate the authors responses to my request to for clarification regarding the survey tools and data presentation. I still have some problems with the concerns I mentioned in my previous review. The authors responded that "We mistakenly used the word ‘correlation’ in place of ‘regression’. Our study was not aimed at conducting correlations." Correlations and regressions are based on similar math. More importantly this is a cross sectional study based on a population of convenience. The sample was not balanced on key factors such as gender, age, seniority, medical specialty, etc. As such it is subject to survival biases and multicollinearity among the study variables. Cross-sectional study can show associations among variables, but they don't establish causation. This does not mean that such studies are not useful for building models, but it is important to consider all of the ways that variables not fully controlled in the study can misinform the user. These factors merit a careful discuss and a carefully worded conclusion. I don't believe the discussion between lines 358 and 366 really do this justice. Also, the oblique reference to "loop holes" does little to inform the reader.

Author’s response: Agreed! Prior to conducting regression analyses, we conducted Pearson’s correlation so as to test the regressions’ assumption that independent and dependent variables correlate with a correlation coefficient value of greater than 0.3. We also conducted several other analyses to test regression model’s assumptions such as normality test, Cooks distance, model of fit, correlations (i.e. to address multicollinearity) and ANOVA. We have rephrased our analyses to elaborate further. We have also rephrased the discussion part on the section of study limitation (i.e. “loop holes”) to further include social desirability biases. 

2) Finally, I find the conclusions overreaching. " We believe that ... we encourage ... We encourage... We also call..." This is a call for action -- not a conclusion. This is a scientific paper -- please stick to the facts.

Authors’ response: Agreed: We have rephrased the conclusion section.

Reviewer #2: 

Burnout and occupational stress are a very important topic these days. Studies on the relationship between self-esteem and Burnout is very interesting to prevent this syndrome in nurses. Some points were not clear in the research and can be improved.

1- The conclusion is very small and it was not clear whether the objectives have been achieved. If the objectives were achieved or not it has been on course completion.

Authors’ response: Agreed: We have rephrased the conclusion section.

2- Krause, N., R. Rugulies, and C. Maslach They are in the references, but are not mentioned in the main text. Please, Check it.

Authors’ response: This reference is cited in the line 67, as number 7.

3- The authors report that there is little research on this topic, but do not suggest further studies on the subject. It is important to suggest to readers new research on the topic, becouse the authors reports there are few studies on the subject.

Authors’ response: Agreed! We have called upon robust research on the topic, by adding a sentence in the discussion section (Line 372).

Reviewer #3: 

Thank you for your responses to my review comments. I believe your study is interesting. However, I still have major concerns related to the current version of this manuscript. Therefore, I would like to suggest some further considerations, revisions and major points of reflections:

Main concerning points:

1. To test the relationship between self-esteem and burnout, the authors report having conducted only regression analyses in their study. A correlation matrix is not reported or mentioned. A correlation test is essential for running regressions.

Authors’ response: Agreed! Prior to conducting regression analyses, we conducted Pearson’s correlation so as to test the regressions’ assumption that independent and dependent variables correlate with a correlation coefficient value of greater than 0.3. We also conducted several other analyses to test regression model’s assumptions (i.e. normality test, Cooks distance, model of fit, correlations and ANOVA). We re-run our analysis and rephrased our analyses to elaborate further. 

2. The authors report the regressions with a ‘correlation coefficient’, and regressions are presented in the form of correlations ‘the higher the self-esteem, the lower level of emotional exhaustion’. This is incorrect and may lead to think that regressions have been conducted without running correlations analyses. If the authors aimed to test the association between these two variables, then they should re-run the analyses because the statistical test selected is not correct.

Authors’ response: Agreed! Prior to conducting regression analyses, we conducted Pearson’s correlation so as to test the regressions’ assumption that independent and dependent variables correlate with a correlation coefficient value of greater than 0.3. We also conducted several other analyses to test regression model’s assumptions (i.e. normality test, Cooks distance, model of fit, correlations and ANOVA). We re-run our analysis and rephrased our analyses to elaborate further. 

3. Another primary concern comes from the number of participants excluded. In one of the points under review, the authors suggested that a large number of participants was excluded ‘due to fear that information was collected by government’. This is an important point that should be included in the study. Indeed, the fact that a large percentage of the recruited participants decided not to take part may lead to bias (e.g. social desirability?). Also, it seems that the percentage of nurses reporting higher self-esteem is much higher than in higher countries. May this be related to, again, this bias?

Authors’ response: Agreed! We have added this in the descriptive data (Line 215); as well as in the discussion part (Line 325). 

4. The manuscript needs proof-reading throughout. There are still severe shortcomings (e.g. two styles of referencing style in the Discussion).

Authors’ response: Agreed! We have standardized our citations in the discussion section.

Previous points

#1. Abstract. Length. In my opinion, this abstract remains still pretty long. I would recommend the following changes.

-line 26: remove ‘using linear regression’ and ‘using ordinal logistic regression the objectives as these are part of the method of analyses

-line 32: remove ‘ Demographic characteristics were recorded and descriptive statistics were illustrated.’

-line 35: change ‘by SPSS’ to ‘by utilizing SPSS’

-line 52-54: remove ‘we also.. to favorability’.

- the conclusion section could be further shortened

Authors’ response: Agreed! We have rephrased the abstract to shorten it.

#8. Methods. Exclusion criteria. Thank you for clarifying that issue related to the exclusion criteria. However, if the nurses agreed to take part only if they did not have fear to be fired by the government, I reckon that your results may suffer from social desirability bias? I still think you should add more specification to the exclusion criteria in the Method section. Also, I would add this point as a limitation in the Discussion.

Authors’ response: Agreed! We have added this factor to the exclusion criteria; and in the discussion section as bias.

New points

2. Introduction. I am glad the authors revised some sections. However, I think that the Introduction lacks a good and strong rationale, especially for self-esteem, effort-reward and work environment. There is a mention of the rationale for burnout but not for the other variables. Also, every construct needs an introduction. Therefore, I would suggest the following structure: 1) introduction of the construct; 2) definition, 3) rationale, 4) description of the Measures. In the current version of the manuscript, the authors defined burnout, then introduced the authors' variables and came back to the definition of burnout with a definition. I would recommend changes in this structure. Also, it would be good to mention more supporting literature.

Authors’ response: Agreed! We have rephrased the entire introduction section.

2b. Introduction. Line 79, you mentioned a large cross-sectional study, there is no need to mention the percentage of every country.

Authors’ response: Agreed! We have rephrased the entire introduction section.

3. Participants. From Line 127 ‘Furthermore...’ you include information related to the Procedure and not the Participants of the study. This info should be placed either into a new section named ‘Procedure’.

Authors’ response: Agreed! We have rephrased accordingly.

4. Data measurements and Instruments. I would place this section before variables. I would include information about the Variables under the description of each questionnaire in the ‘Data measurements and Instruments’.

Authors’ response: Agreed! We have rephrased accordingly.

5. Analyses. Correlations. The authors report the regressions but do not report the correlation matrix. I don’t think it is acceptable to undertake regressions without running and reporting a correlation matrix. I strongly encourage the authors to do so.

Authors’ response: Agreed! Prior to conducting regression analyses, we conducted Pearson’s correlation so as to test the regressions’ assumption that independent and dependent variables correlate with a correlation coefficient value of greater than 0.3. We also conducted several other analyses to test regression model’s assumptions (i.e. normality test, Cooks distance, model of fit, correlations and ANOVA). We have rephrased our analyses to elaborate further. 

6. Appendix, Tables and Figures. There are a lot of Appendix, Tables and Figures in the manuscript. First, I would place all the questionnaires into one Appendix. Second, I would reduce the number of Tables and Figures. Information placed in-text does not necessarily need to be repeated elsewhere.

Authors’ response: Agreed! We have deleted the 4 figures and Table 5 which previously illustrated coefficients. We merged all questionnaires into one appendix, Appendix 1.

7. Ethical consideration. This information is usually placed at the start of the Method section (e.g. under ‘Participants’).

Authors’ response: Agreed! We have rephrased accordingly.

8. Results. Information placed under the Descriptive Data regarding nurses should go under Participants.

Authors’ response: Agreed! We have rephrased accordingly.

9. The ‘work environment’ construct may be easily confused with work stressors (organizational stressors). I would recommend referring to this variable as ‘favourability to the work environment (context)’.

Authors’ response: Agreed! We have rephrased accordingly.

10. Discussion. References. The Discussion section still includes two types of referencing styles. The authors should only include ‘Vancouver’ as the preferred referencing style.

Authors’ response: Agreed! We have standardized the references in the discussion part. However, Journal’s’ instructions to authors recommend PLoS One style which can be downloaded from the website. 

11. Discussion. I am pretty impressed to read that 68% of the nurses have high self-esteem. Is this related to the social desirability bias and the fact that nurses felt the pressure from the government to respond? If you think that may be the case, it would be good to mention it.

Authors’ response: Agreed! We have rephrased the discussion part to include social desirability bias.

Authors are indeed grateful for your time reviewing our manuscript and promptly respond. Kindly, review our manuscript and hopefully publish. I am looking forward hearing from you soon.

Best regards,

Yasira kabakleh, BScN

Department of Nursing

Xiangya Nursing School, 

Central South University,

172 Tongzipo Road, Yuelu District, 

Changsha City, P.R. China 410013

Mobile: +8615675802994

Email: yasiraghazi@gmail.com

---

## [Decision Letter · Decision Letter 2]

24 Aug 2020

Burnout and associated occupational stresses among Chinese nurses: A cross-sectional study in three hospitals

PONE-D-19-33567R2

Dear Dr. Zhang,

We’re pleased to inform you that your manuscript has been judged scientifically suitable for publication and will be formally accepted for publication once it meets all outstanding technical requirements.

Kind regards,

Sergio A. Useche, Ph.D.

Academic Editor

PLOS ONE

Additional Editor Comments (optional):

Reviewers' comments:

Reviewer's Responses to Questions

**Comments to the Author**

1. If the authors have adequately addressed your comments raised in a previous round of review and you feel that this manuscript is now acceptable for publication, you may indicate that here to bypass the “Comments to the Author” section, enter your conflict of interest statement in the “Confidential to Editor” section, and submit your "Accept" recommendation.

Reviewer #1: All comments have been addressed

2. Is the manuscript technically sound, and do the data support the conclusions?

Reviewer #1: Yes

3. Has the statistical analysis been performed appropriately and rigorously? 

Reviewer #1: Yes

4. Have the authors made all data underlying the findings in their manuscript fully available?

Reviewer #1: Yes

5. Is the manuscript presented in an intelligible fashion and written in standard English?

Reviewer #1: Yes

6. Review Comments to the Author

Reviewer #1: I have no further comments. This paper addresses an important topic and my be of use to practitioners and some researchers.

7. PLOS authors have the option to publish the peer review history of their article (what does this mean?). If published, this will include your full peer review and any attached files.

Reviewer #1: No

---

## [Editor Report · Acceptance letter]

28 Aug 2020

PONE-D-19-33567R2 

Burnout and associated occupational stresses among Chinese nurses: A cross-sectional study in three hospitals 

Dear Dr. Zhang:

I'm pleased to inform you that your manuscript has been deemed suitable for publication in PLOS ONE. Congratulations! Your manuscript is now with our production department. 

Kind regards, 

on behalf of

Dr. Sergio A. Useche 

Academic Editor

PLOS ONE